# Evaluation of Recurrent Disease after Radiation Therapy for Patients Considering Local Salvage Therapy: Past vs. Contemporary Management

**DOI:** 10.3390/cancers15245883

**Published:** 2023-12-18

**Authors:** Eric S. Adams, Sriram Deivasigamani, Mahdi Mottaghi, Jiaoti Huang, Rajan T. Gupta, Thomas J. Polascik

**Affiliations:** 1Department of Urology, Duke University Medical Center, Durham, NC 27710, USA; 2Section of Urology, Department of Surgery, Durham Veterans Affairs Medical Center, Durham, NC 27710, USA; 3Department of Pathology, Duke University Medical Center, Durham, NC 27710, USA; 4Department of Radiology, Duke University Medical Center, Durham, NC 27710, USA

**Keywords:** prostate cancer, localized recurrence, radiorecurrent, radiotherapy failure, radiation therapy failure, patient evaluation, patient selection, salvage therapy

## Abstract

**Simple Summary:**

Recurrence of prostate cancer after radiation therapy is a common clinical scenario. Salvage therapy can be beneficial for some patients with localized recurrence, but careful patient evaluation is important to determine their suitability and ensure an appropriate salvage treatment strategy is selected. This review provides a comprehensive overview of the evaluation of patients with recurrent prostate cancer following radiation therapy, focusing on how the evaluation of these patients has continued to evolve over time.

**Abstract:**

Recurrent prostate cancer after primary treatment with radiation therapy is a common problem. Patients with localized recurrence may benefit from salvage therapy, but careful patient selection is crucial because not all patients will benefit from local salvage therapy, and salvage therapy has increased morbidity compared to primary treatments for prostate cancer. This review aims to provide an overview of the evaluation of patients with recurrent disease after radiation therapy and how it is continuing to evolve with increasing data on outcomes, as well as improving technologies and techniques. Our enhanced understanding of treatment outcomes and risk stratification has influenced the identification of patients who may benefit from local salvage treatment. Advances in imaging and biopsy techniques have enhanced the accuracy of locating the recurrence, which affects treatment decisions. Additionally, the growing interest in image-targeted ablative therapies that have less morbidity and complications than whole-gland therapies for suitable patients influences the evaluation process for those considering focal salvage therapy. Although significant changes have been made in the diagnostic evaluation of patients with recurrent disease after radiation therapy, it remains unclear whether these changes will ultimately improve patient outcomes.

## 1. Introduction

Prostate cancer (PCa) is the most common solid-organ cancer, and second leading cause of cancer death in men in the United States [1]. Radiation therapy (RT) has been and continues to be utilized for the primary treatment of clinically localized PCa with curative intent using both external beam radiation therapy (EBRT) and brachytherapy (BT). Despite advances in RT, biochemical recurrence (BCR) after primary RT for localized PCa remains common between 15 and 57% [2]. Although BCR does not necessarily imply a dismal prognosis, it is particularly important in intermediate- and high-risk patients, as up to one-third of these patients may progress to metastasis and die of PCa [3]. In the past decades, up to 90% of patients with radiorecurrent prostate cancer have received palliative androgen deprivation therapy (ADT) instead of local salvage treatment [4]. This exposes patients to the side effects and decreased the quality of life associated with ADT and does not offer these patients a chance for a definitive cure.

For patients with localized radiorecurrent PCa, local salvage therapy offers patients an opportunity of a definitive cure and potential to avoid the side effects associated with systemic therapies and ADT [2,5,6,7]. Multiple local salvage therapies are available for patients with radiorecurrent PCa, including salvage radical prostatectomy (sRP), salvage ablative therapies (including cryoablation and high-intensity focused ultrasound (HIFU), and repeat RT with stereotactic body radiotherapy (SBRT), brachytherapy (BT) or EBRT [7,8]. Patients with radiorecurrent PCa who are being considered for local salvage therapy should undergo an assessment of their life expectancy and Pca risk, including restaging their PCa. An accurate restaging of patients with radiorecurrent PCa via imaging and biopsies is a critically important component of patient selection. Fortunately, advancements in imaging, including multiparametric prostate MRI (mpMRI) and the use of nuclear medicine radiotracers, has greatly improved accuracy in the detection and localization of PCa recurrence. More accurate restaging of patients with radiorecurrent PCa may give clinicians and patients greater confidence that patients who choose to undergo local salvage therapy would potentially be cured. However, it is yet to be determined whether this translates to better clinical outcomes for patients and how patients who are found to have a small-volume metastasis could optimally be managed.

The decision of whether to pursue local salvage therapy, whether to consider additional metastasis-directed therapy, and which salvage technique(s) to pursue should be an individualized, shared decision between the patient and their clinician(s). Advancements in local salvage therapies including cryoablation, HIFU, SBRT, and BT resulted in acceptable short-to-mid-term oncological and favorable functional outcomes in patients with localized radiorecurrent PCa; however, salvage treatment is still associated with higher risks of complications and worse quality of life outcomes than primary treatment [6,7,8]. Therefore, given these higher risks and toxicities associated with salvage therapy, patient selection for local salvage therapy is critically important.

Comparisons between available salvage techniques are challenging due to the lack of prospective and randomized studies. Nonetheless, other systematic reviews have attempted to make comparisons between salvage techniques for radiorecurrent PCa using available evidence [8,9]. However, in addition to heterogeneous data, these reviews did not differentiate between whole-gland therapies and focal therapies, the latter of which, for appropriate patients, appears to offer similar oncologic control but with fewer complications and less morbidity [2,10,11,12]. Comparisons between salvage therapy options is beyond the scope of this paper; instead, we focus on important changes in the diagnostic evaluation of patients considering local salvage therapy that are used for patients considering any local salvage therapy. Herein, we aim to review the evaluation of patients with radiorecurrent PCa who are considering local salvage therapies and how the evaluation of these patients has changed over the past decades. Important changes include evolution in definitions and risk stratification of patients with RT failure, as well as changes in restaging with new imaging and biopsy techniques.

## 2. Defining Biochemical Recurrence after Primary Radiation Therapy for Prostate Cancer

During surgery utilizing RP, the surgeon will attempt to remove the entire prostate gland, so when the surgery is a biochemical success no prostate tissue will remain and the patient will not have a detectable PSA. For patients who undergo RP, a relatively low and verified PSA level of ≥0.2 ng/mL has been used by the American Urological Association (AUA) to define BCR [13]. After RP, a relatively low PSA level of >0.4 ng/mL that is continuing to rise is predictive of future metastasis [14]. Using PSA monitoring after RT to define the success or failure of treatment and biochemical recurrence is less straight forward. This is because RT causes DNA damage that preferentially causes cellular death in tumor cells compared to benign prostate cells due to impaired DNA repair mechanisms in cancer. Therefore, benign prostate tissue can remain and produce PSA even when RT is successful at treating the patient’s cancer, since RT preferentially kills cancer cells.

There is no agreed upon PSA nadir value that has been used to define the success or failure of RT with EBRT or brachytherapy. A PSA nadir value of ≤0.2 ng/mL at 4 years post-RT with brachytherapy has been proposed as a potential definition for a “biochemical cure” because it is highly predictive of disease-free survival at 10 years [15,16]. However, a higher post-RT nadir PSA value cannot be considered a treatment failure because a higher PSA nadir is not sufficiently predictive of cancer progression. Defining BCR following RT is more challenging than after RP. Part of this is because of the phenomenon of a PSA bounce, particularly after brachytherapy, a small rise in PSA after the completion of RT that is common and physiologic [17]. As such, several definitions for BCR after RT have been proposed over the years.

In 1996 the “ASTRO definition” of biochemical failure was determined by a consensus conference to be three consecutive rises in PSA after the patient’s PSA reaches a post-RT nadir [18,19]. However, due to concerns that the ASTRO definition was too strict and did not sufficiently predict clinical outcomes such as clinical progression and cancer-specific survival, the “Phoenix criteria” arose from a subsequent consensus conference in 2005. The Phoenix criteria defined BCR after RT (with or without short-term ADT) as a PSA value ≥2 ng/mL above the patient’s post-RT nadir PSA; however, they also recommended considering treatment failure if a patient underwent salvage therapy at the time of a positive biopsy or initiation of salvage therapy (whichever comes first) [19]. The Phoenix criteria of BCR after RT is still commonly used in clinical practice, as well as current guidelines in the US and Europe [13,20].

A “one-size-fits-all” definition of BCR such as the Phoenix criteria may be important for research purposes, but its limitations must be recognized when treating individual patients. One of the major problems with using the Phoenix criteria is that waiting until a patient’s PSA reaches the threshold of 2 ng/mL above nadir can delay salvage treatment for patients whose PSA is rising, indicating that their cancer may be progressing. Since modern imaging techniques can be used to detect PCa recurrence below the threshold of the Phoenix criteria, some authors have suggested that the Phoenix criteria should be reevaluated [21]. For example, one study suggested that patients who undergo EBRT with a BT boost could utilize the same definition of BCR as patients who underwent RP [22]. Delaying salvage treatment for patients until their PSA reaches the Phoenix criteria cutpoint decreases the likelihood that local salvage therapy is curative because it increases the time that patients’ cancer has to metastasize prior to the salvage treatment. Therefore, select patients who have a continuing rise in PSA after RT and who are potential candidates for local salvage therapy should undergo evaluation prior to reaching the Phoenix criteria threshold of 2 ng/mL above their nadir PSA level.

## 3. Initial Risk Stratification of Patients with Biochemical Recurrence after Primary Radiation Therapy

Many patients who have BCR after RT will not have clinically significant recurrence of PCa or death from PCa, so not all patients with BCR after RT warrant salvage with local or systemic therapy [13]. The first step in determining if a patient with suspected BCR after RT should undergo early evaluation for consideration of possible local salvage treatment is an evaluation of their life expectancy, comorbidities, and preferences [13]. In a relatively large cohort of patients who underwent EBRT and experienced BCR, the median time to PCa-specific death after recurrence was 10 years [23]. If patients are unlikely to benefit from salvage therapy due to a limited life expectancy, then there is not an urgency for them to undergo early evaluation for potential local salvage therapy. Similarly, if patients are not interested in pursuing local salvage therapy if they are determined to be a candidate, then an early work-up to assess whether they are a candidate is similarly not necessary. For patients who may benefit from and desire local salvage therapy, an early evaluation should be considered for select patients based on their risk of PCa progression.

There are several clinical factors which have been demonstrated in numerous studies to correspond to a higher risk of PCa progression for patients who have undergone primary treatment with RT. Those risk factors are a higher Gleason score prior to primary therapy, shorter interval to biochemical failure, and shorter PSA doubling time (PSA-DT) [14]. Additional risk factors which have been demonstrated in some studies include a higher clinical T stage and higher PSA value prior to primary treatment [14], as well as a higher nadir PSA value after primary RT [24].

The European Association of Urology (EAU) proposed a risk stratification system for patients with BCR after primary RT based on the patient’s tumor grade prior to primary treatment and the interval from treatment to biochemical failure [25]. In their proposed risk stratification system, patients are considered high-risk BCR if they had an ISUP grade group 4 or 5 prior to RT, or they had an interval from nadir to BCR of ≤18 months, whereas patients with a lower-grade PCa prior to treatment and longer interval to BCR are considered low risk [25]. This risk stratification system can identify patients with low-risk BCR who can be offered close surveillance and delayed salvage therapy. However, this risk stratification system does not resolve the problem of delayed work-up and treatment for patients with high risk who have not yet reached the Phoenix criteria of BCR.

An alternative method of risk stratification was proposed by Zumsteg et al. based on the identification of patients’ risk factors for PCa progression following BCR after RT [23]. These risk factors include preRT ISUP grade group 4 or 5, preRT clinical stage cT3b-T4, interval from nadir to BCR of <3 years, and post-RT PSA-DT < 3 months [23]. Zumsteg’s et al. risk factors overlap with the EAUs discussed above, but because it includes PSA-DT as a predictive risk factor, it provides insight to interpret a patient’s rising PSA before it has reached the Phoenix criteria threshold of BCR. Select patients who are identified as higher-risk due to Gleason score prior to RT and/or PSA-DT after RT warrant evaluation for possible local salvage therapy prior to reaching the Phoenix criteria of BCR, such as the previously used ASCO definition of three consecutive increases in PSA. Table 1 provides an overview of the risk stratification systems.

## 4. Imaging Evaluation for the Restaging of Patients with Prostate Cancer Recurrence after Primary Radiation Therapy

### 4.1. Conventional Imaging with Computed Tomography Scan and Bone Scan

Following suspicion of cancer persistence after RT and after meeting the biochemical definition of recurrence, imaging is used to re-stage the disease. The past standard of care imaging for restaging PCa in the setting of BCR included computed tomography (CT) scan and a whole-body technicium bone scan [13,26]. However, these imaging modalities have limited sensitivity to detect metastatic PCa early in the progression of metastatic PCa and at low PSA levels [27]. Previous studies have demonstrated that conventional imaging with CT and bone scans have a very low yield at a PSA level less than 10 ng/mL [27]. Bone scans also have the challenge of limited specificity with difficulty differentiating between bone metastasis and other pathological conditions such as degenerative bone disease that is common in the PCa demographic [28]. The use of single-photon emission computed tomography/computed tomography (SPECT/CT) improves diagnostic confidence and increases specificity compared to traditional planar bone scans, but does not overcome the limited sensitivity of bone scans early in the progression of mPCa [28].

### 4.2. Multiparametric Magnetic Resonance Imaging of the Prostate

Current multiparametric magnetic resonance imaging (mpMRI) of the prostate combines T2-weighted imaging (T2WI) with diffusion-weighted imaging (DWI) and dynamic contrast-enhanced imaging (DCE-MRI) to improve intraprostatic tissue resolution and diagnostic sensitivity [29]. The utilization of mpMRI of the prostate improves sensitivity for local PCa detection compared to T2WI of the prostate alone [29,30]. This is particularly important in restaging of patients who have undergone RT for PCa, because both recurrent PCa and irradiated PCa can appear hypointense on T2WI [31]. The first study to demonstrate the added value of contrast-enhanced magnetic resonance imaging (MRI) for the localization of recurrent PCa after primary RT treatment was published in 2014 [32].

The use of mpMRI can aid in the detection of locally recurrent PCa after RT and quantification of local PCa extension for use in salvage treatment planning [33]. Although mpMRI has good spatial resolution, it has been demonstrated to underquantify and under-stage radiorecurrent PCa [34]. Further, it can be challenging to detect recurrence in the field of brachytherapy implants. A recent meta-analysis of mpMRI of radiorecurrent PCa demonstrated that mpMRI has high specificity but limited sensitivity for local and nodal radiorecurrent PCa [35]. The use of novel positron emission tomography (PET) agents in combination with mpMRI has been shown to improve the sensitivity, specificity, and localization for primary PCa [36,37,38]. There are limited studies evaluating the combination of mpMRI and the newer prostate-specific PET radiotracers in the setting of radiorecurrent PCa, but one such study found that when patients had suspicion of radiorecurrent PCa on both mpMRI and novel PET imaging, the positive predictive value for confirmation on biopsy histology was 97.6% [39].

### 4.3. Positron Emission Tomography with Novel Radiopharmaceuticals

The use of ^18^F-fludeoxyglucose (FDG) positron emission tomography (PET) scans is limited in the setting of PCa due to the limited uptake of FDG in prostate cancer [40]. The introduction of novel radiopharmaceuticals has continued to improve the sensitivity of PET scans to more accurately stage and restage PCa than previously possible. PET-CT or PET-MRI can be used as a single imaging evaluation study instead of two studies, with conventional imaging utilizing both a bone scan and CT scan or MRI performed separately, since PET-CT or PET-MRI can evaluate for bone and soft tissue metastasis concomitantly. Current guidelines include the use of novel PET-CT scans in patients after failure of local therapy either as an alternative to conventional imaging or if conventional imaging is negative [13].

The first radiolabeled tracers introduced for use in PCa imaging were based on acetate [41], choline [42], and fluciclovine [43]. Although the use of these tracers has been increasingly common, they still have limited cancer detection at low PSA levels and limited specificity [44]. However, more recently, there has been growing development and use of radiopharmaceuticals that specifically target prostate-specific membrane antigen (PSMA). PSMA-targeted PET scans have been shown to have better detection rates than PET scans using other radiolabeled tracers, including fluciclovine [43,45,46]. However, PSMA is not exclusive to prostate tissue despite the term “prostate-specific” in its name, and the PSMA ligand uptake can be noted in benign tissue including healing bone fractures and fibrocartilage lesions [47].

The cancer detection rate of all novel PET scan techniques is dependent on the patient’s PSA level, but PSMA PET is the most sensitive and has demonstrated positive findings in 42% of patients with a PSA level below 0.2 ng/mL after RP in a pooled analysis [48]. Therefore, the selection of a novel PET agent and scan technique to utilize to restage a patient after RT may depend on availability and insurance coverage, but for patients with a low PSA, the use of PSMA PET should be preferred. The use of PSMA PET scans should not be limited to patients who have met the Phoenix criteria of BCR after RT, since PSMA PET scans can be used to localize recurrent PCa at low PSA levels, and local salvage therapy can be initiated at PSA levels below the Phoenix criteria [44]. When a panel of international prostate cancer experts were asked at what PSA level they recommend imaging for asymptomatic rising PSA after primary RT with pre-RT biopsy ISUP grade group 4–5 or interval to biochemical failure <18 months, 38% of experts reported performing imaging before PSA reaches the Phoenix criteria of BCR after RT [49].

### 4.4. Changes in Management and Controversies Resulting from Increased Sensitivity of Next-Generation Imaging

The use of next-generation imaging techniques for prostate cancer restaging can now diagnose low-volume metastatic PCa or “oligometastatic” PCa in patients who previously would have had negative conventional imaging and previously would have been managed as having clinically localized disease. There are several studies that have shown next-generation imaging leads to a change in the management of patients undergoing evaluation for BCR after RT and RP [50,51,52,53]. There is hope that the use of highly sensitive PET scans for treatment planning will improve outcomes for patients with BCR after RT, but there are yet to be any completed clinical trials assessing this [54].

When patients are found to have low volume evidence of metastasis that is only found on novel PET imaging, clinicians must first assess whether the finding requires further work-up. Because novel PET imaging has a risk of false-positive findings, particularly due to nonspecific uptake in the bone, patients with a metastasis detected only on PET imaging should have an individualized decision made on whether to consider the PET scan findings as accurate or perform further work-up of the potential metastasis with additional imaging and/or biopsy [49]. If the finding of low-volume metastasis found on novel PET imaging is accurate, clinicians must then assess whether this prevents patients from being candidates who may benefit from local salvage therapy.

In the past, if patients were found to have metastasis on conventional imaging or lymph node dissection, they were not considered candidates for local salvage therapy. However, the increased sensitivity of next-generation imaging that exceeds conventional imaging has added complexity and controversy in deciding how to manage patients with local recurrence and low-volume metastatic PCa. Unfortunately, there is currently a lack of evidence on how to best manage patients with low-volume metastatic PCa found only on novel PET imaging in both the primary and salvage treatment settings. This leaves unanswered questions as to whether these patients should still have local salvage therapy as they would have if they had only undergone restaging with conventional imaging. Should they undergo local salvage therapy with the addition of metastasis-directed therapy, or should they only be considered for systemic therapy?

There is some suggestion of a possible benefit of local prostate treatment for patients with low-volume metastatic PCa even if the metastases are not directly treated, but this is not part of standard of care treatment as of yet [55,56]. Additionally, the use of metastasis-directed therapy has been suggested as a treatment strategy for patients with low-volume metastasis. A small number of phase II trials have supported the potential role of metastasis-directed therapy for patients with low-volume metastatic PCa [57,58,59]. Thus, patients with local recurrence and low-volume metastasis may benefit from a combination of local salvage therapy and metastasis-directed therapy; however, further research is needed to determine who is most likely to benefit from this treatment strategy.

At present the decision on how to manage patients with low-volume metastasis found on novel PET imaging and whether this rules out local salvage therapy is a complex and controversial issue. We suggest that this is best addressed with a multi-disciplinary team using an individualized approach involving the input of the patient and their preferences. However, the important take away is that unlike prior management where patients with metastasis were not considered candidates for local salvage therapy, select patients with metastasis on novel PET imaging may still be candidates for local salvage therapy.

## 5. Biopsy Evaluation of Patients with Suspected Local Prostate Cancer Recurrence after Primary Radiation Therapy

### 5.1. Indication for Prostate Biopsy after Primary Radiation Therapy

A positive prostate biopsy after primary RT for PCa is predictive of progression to metastasis and worse cancer-specific survival [60,61]. However, post-RT biopsy has some limitations and is not routinely performed to assess for treatment failure unless there is biochemical evidence of treatment failure [62]. Prostate biopsy in this setting aims to detect local recurrence/persistence of prostate cancer and evaluate the grade and extent of local extent [63]. However, because RT influences the prostate’s histologic morphology and can kill tumor in a delayed fashion, this limits the ability to pathologically grade tumors with a significant radiation effect [64].

### 5.2. The Risk of False-Positive and Indeterminate Prostate Biopsy after Radiation Therapy

The pathologic evaluation of prostate tissue after radiation becomes more challenging and can result in indeterminate biopsies due to residual tumor with radiation effects of uncertain viability and false positives due to delayed tumor regression [63]. Indeterminant biopsy results are common shortly after radiation therapy, and can be found in up to 40% of post-radiation biopsies [65]. As the interval of time from RT to post-radiation biopsy increases, the frequency of indeterminate biopsy decreases, so some authors suggested delaying post-radiation biopsy for a longer interval after completion of RT [66]. Similarly, multiple studies have demonstrated that the cytotoxic effects of radiotherapy can result in a delayed resolution of positive biopsy findings, taking even up to 30–36 months following RT [67,68,69]. However, the patients who experience early BCR are at the highest risk for PCa progression and would be most likely to benefit from early salvage therapy [23,25], so post-radiation biopsy should not be delayed for these patients despite the risk of indeterminate biopsy or a false positive.

There is uncertainty regarding how we should interpret post-radiation biopsy with indeterminate results due to tumor with radiation treatment effect. One study found that patients with biopsies categorized as indeterminate due to residual tumor with severe radiation treatment effect had a similar clinical progression rate to patients with negative biopsies [70]. However, a different study found that on repeat biopsies of patients with indeterminate biopsy, 18% of patients progressed to local treatment failure [66].

### 5.3. The Risk of a False Negative and Under-Staging on Prostate Biopsy after Radiation Therapy

In addition to post-radiation biopsies being at risk of a false positive and indeterminate results, post-radiation biopsies are also prone to false-negative results and under-staging due to insufficient sampling [63]. A comparison of biopsy pathology to surgical pathology for patients undergoing salvage RP following RT showed 58% of patients had upgrading of their pathology after surgery and up to 20% of tumors were missed [71].

A negative post-radiation biopsy in the presence of BCR and absence of metastasis on imaging should not provide complete reassurance and should warrant further monitoring or workup. Upon an assessment of 286 men with BCR after RT but negative post-radiation prostate biopsy, during a 66-month average follow-up period, 43% of the patients developed metastasis and 15% died of PCa [72]. Furthermore, these authors demonstrated that a negative biopsy did not rule out local recurrence, as 5 of the 9 patients who underwent a repeat biopsy had a positive repeat biopsy [72]. One study suggested the possibility of avoiding the necessity of a biopsy when both mpMRI and PSMA PET are positive [39]; however, further studies are necessary before this can be routinely implemented in clinical practice. Given the morbidity of salvage therapy for locally recurrent PCa after RT, guidelines for patients considering local salvage therapy state that patients should undergo histologic confirmation of local recurrence with prostate biopsy prior to the consideration of treatment [20,49].

### 5.4. Selection of Prostate Biopsy Technique after Primary Radiation Therapy

Multiple techniques are available for post-radiation biopsy with varying accuracy in detecting and staging local PCa recurrence. Select patients undergoing post-radiation biopsy may be better served with transperineal biopsy instead of a transrectal biopsy if they have post-radiation proctitis or history of post-RT rectorrhagia [63], but most patients can undergo a post-radiation prostate biopsy using either approach. Typical prostate biopsies for both transrectal and transperineal approaches use systematic sampling using a template of biopsies [73]. Similar templates can be used for patients undergoing post-radiation biopsy to assess for PCa recurrence, but some adjustments should be made based on the knowledge of common PCa recurrence patterns after RT and for treatment planning. An assessment of patterns of local PCa recurrence post-RT has suggested that post-radiation biopsy should include biopsies of the seminal vesicles, distal apex, and areas adjacent to the urethra [74]. As compared to typical 12-core systematic sampling template, this involves the addition of biopsy cores from both the bases of the seminal vesicles and the distal ends of the seminal vesicles, as well as adjusting the aiming of the template biopsies to ensure the apical biopsies include the distal apex and medial biopsies are closer to the urethra than typical sampling.

The improvements in imaging for detecting PCa have made the use of image-targeted biopsies possible as a promising addition to standard-template biopsies to improve biopsy accuracy [75,76,77]. Imaging-targeted biopsies can be performed based on mpMRI imaging [75,76], novel PET/CT [76], or novel PET/MRI [77]. For patients with suspected local recurrence of PCa after RT and a target on imaging, imaging-guided biopsies have been shown to have high rates of cancer detection [39,52,78]. However, not all patients with locally recurrent PCa have a visible target on imaging, so image-targeted biopsies are not always an option.

Another technique for post-radiation biopsies to consider is three-dimensional template mapping biopsies (3D-TMB), which increases biopsy accuracy without requiring a visible target on imaging. Because 3D-TMB entails more thorough sampling of the prostate, this helps overcome some of the limitations of post-radiation biopsy including the risks of indeterminate biopsy, false negative biopsy, and under-staging [78,79,80]. The use of prostate mapping biopsies is more sensitive than imaging to detect localized PCa recurrence, especially for smaller tumors; thus, it can detect local recurrence of PCa that is not visualized on imaging [78,81]. However, obtaining many more biopsy cores for mapping biopsies increases the morbidity and rate of adverse events associated with prostate biopsies compared to prostate biopsy techniques that require obtaining fewer biopsy cores [82]. Thus, clinicians may consider reserving the use of mapping biopsies for patients without a target on imaging and/or patients with an initial post-radiation biopsy that is negative.

The selection of prostate biopsy technique may also depend on whether the patient is being evaluated for salvage using a focal therapy technique. For patients who are only being considered for locoregional and whole gland salvage therapies the role of biopsy is mainly to confirm the patient’s diagnosis of local recurrence and risk stratify the patient to confirm the need for salvage therapy. However, there has been increasing interest in the use of focal therapies for post-RT salvage focal re-irradiation and focal ablation due to decreased morbidity and toxicity of focal therapies [2,10,11,12]. For patients being considered for focal therapy the location and extent of PCa recurrence must be accurately identified to ensure appropriate patient selection and treatment planning. Therefore, for patients who are only being considered for whole-gland salvage options a standard template biopsy with additional biopsies of the seminal vesicles may be sufficient. However, for patients who are being considered for local salvage with focal therapy, patients should additionally undergo image-guided biopsies, or mapping biopsies if a target is not visualized on imaging. Figure 1 shows the patient evaluation flowchart following radio-recurrence.

## 6. Conclusions

Patients with localized recurrence of PCa after primary RT may benefit from salvage therapy because therapy may be a curative; however, appropriate patient evaluation and selection is critical because not all patients will benefit from salvage therapy, and salvage treatment has increased morbidity compared to primary treatment for PCa. The monitoring of PSA after primary RT has been used to define treatment failure, but there is a lack of consensus on what PSA threshold and kinetics should trigger patients to be evaluated for possible salvage therapy with restaging imaging and histologic biopsy confirmation. Increased studies on outcomes for patients with BCR after RT has suggested that only patients with high-risk BCR and sufficient life expectancy will likely benefit from salvage therapy.

Evaluation of patients who may benefit from local salvage therapy under contemporary management includes restaging imaging followed by confirmatory prostate biopsy including seminal vesicle biopsy. Conventional imaging with a CT scan and bone scan has poor sensitivity for the detection of radiorecurrent PCa, especially at low PSA levels, and have been replaced by mpMRI and novel PET scans that are much more sensitive in detecting radiorecurrent PCa.

Prostate biopsy after radiation therapy has limitations, but histological confirmation of the presence of localized PCa is still necessary prior to the implementation of local salvage treatment for radiorecurrent PCa. Standard templates for systematic prostate biopsies should be adjusted to ensure an adequate sampling of the seminal vesicles, distal apex, and periurethral prostate. Advanced biopsy techniques to improve accuracy when evaluating for local recurrence of PCa after RT include image-targeted biopsies and mapping biopsies.

The use of improved imaging and biopsy techniques has increased the accuracy of the detection and localization of radiorecurrent PCa. This increased information has been influencing treatment decisions for patients considering local salvage therapy for radiorecurrent PCa, although the impact on patient outcomes is still being evaluated. These changes include an increased interest in focal therapy which has been made possible due to the improved localization and targeting of radiorecurrent PCa. Whether these changes in evaluation and the resultant changes to treatment decisions will have a positive impact on patient outcomes is yet to be determined and will require further studies to assess their impact on clinically meaningful patient outcomes.

## Figures and Tables

**Figure 1 cancers-15-05883-f001:**
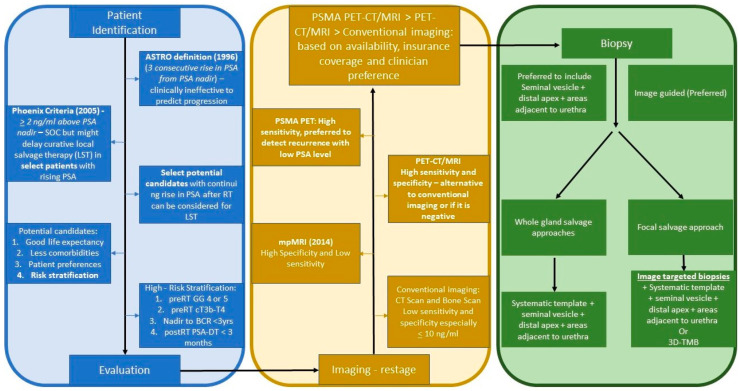
Patient evaluation flowchart following radio-recurrence. Footnotes: PSA, prostate-specific antigen; SOC, standard of care; RT, radiotherapy; GG, Gleason Grade; BCR, biochemical recurrence; PSA-DT, prostate-specific antigen doubling time; CT, computerized tomography; mpMRI, multiparametric magnetic resonance imaging; PET-CT/MRI, positron emission tomography-CT/MRI; PSMA PET, prostate-specific membrane antigen PET; 3D-TMB, three-dimensional transperineal mapping biopsy.

**Table 1 cancers-15-05883-t001:** Risk stratification systems in clinical practice.

Clinical Characteristics	**European Association of Urology (EAU)**	
	EAU low-risk BCR	EAU high-risk BCR
	PSA Doubling Time > 1 year, AND	PSA Doubling Time of ≤1 year OR
	ISUP < 8, AND	ISUP ≥ 8, OR
	IBF of >18 months	IBF of ≤18 months
Clinical Characteristics	**Zumsteg’s Criteria**	
	Unfavorable for Distant Progression *	
	ISUP ≥ 8,	
	Pretreatment stage ≥ cT3b,	
	IBF < 3 years OR PSA-Doubling Time < 3 months	

* Individuals with exactly one risk factor have a lower risk of progression than those with two or more risk factors; ISUP: International Society of Urological Pathologists, IBF: Interval to Biochemical Failure.

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
