# Peer review of "Evaluation of Recurrent Disease after Radiation Therapy for Patients Considering Local Salvage Therapy: Past vs. Contemporary Management"

_cancers, 2023, doi:10.3390/cancers15245883_

Round 1
Reviewer 1 Report
Comments and Suggestions for Authors
This is a well-written review of an emerging issue. Other papers faced this topic but the present one seems to be really interesting and updated.
I strongly suggest to add some sentences evalutating the even more modern possibility to re-irradiate these patients (future perspective) including these papers:
Archer P, et al. Salvage Stereotactic Reirradiation for Local Recurrence in the Prostatic Bed After Prostatectomy: A Retrospective Multicenter Study. Eur Urol Oncol. 2023 doi: 10.1016/j.euo.2023.03.005.
Reviewer 2 Report
Comments and Suggestions for Authors
Introduction:
"Multiple local salvage therapies are available for patients 52 with radiorecurrent PCa including salvage radical prostatectomy (sRP), salvage ablative 53 therapies (including cryoablation and high-intensity focused ultrasound (HIFU), and repeat RT with brachytherapy or EBRT.": in order to give an improved context, I would suggest to cite some recent literature about reirradiation and non surgical local treatments in this field. To improve the background section, I would suggest to refer to PMID: 37728816
Reviewer 3 Report
Comments and Suggestions for Authors
I want to congratulate the authors for their rigorous, well-executed, and structured review work. However, I consider that the inclusion in it of a table that shows the different criteria to evaluate the progression criteria such as Zumsteg's and the other existing ones would facilitate the understanding of the text. I consider that the same could be done with the existing alternatives for selecting the technique for prostate biopsy. I believe these tables will enrich your work.
